# A unified neurocognitive model of semantics language social behaviour and face recognition in semantic dementia

Junhua Ding [1,2,10], Keliang Chen[3,10], Haoming Liu[4], Lin Huang[5], Yan Chen[1,6], Yingru Lv[7], Qing Yang[8], Qihao Guo[5,11✉], Zaizhu Han[1,11✉] & Matthew. A. Lambon Ralph [9,11✉]

The anterior temporal lobes (ATL) have become a key brain region of interest in cognitive neuroscience founded upon neuropsychological investigations of semantic dementia (SD). The purposes of this investigation are to generate a single unified model that captures the known cognitive-behavioural variations in SD and map these to the patients' distribution of frontotemporal atrophy. Here we show that the degree of generalised semantic impairment is related to the patients' total, bilateral ATL atrophy. Verbal production ability is related to total ATL atrophy as well as to the balance of left > right ATL atrophy. Apathy is found to relate positively to the degree of orbitofrontal atrophy. Disinhibition is related to right ATL and orbitofrontal atrophy, and face recognition to right ATL volumes. Rather than positing mutually-exclusive sub-categories, the data-driven model repositions semantics, language, social behaviour and face recognition into a continuous frontotemporal neurocognitive space.

[1] State Key Laboratory of Cognitive Neuroscience and Learning and IDG/McGovern Institute for Brain Research, Beijing Normal University, Beijing, China. [2] Department of Neurosurgery, Baylor College of Medicine, Houston, TX, USA. [3] Department of Neurology, Huashan Hospital, Fudan University, Shanghai, China. [4] Department of Asian and North African Studies, Ca' Foscari University of Venice, Venice, Italy. [5] Department of gerontology, Shanghai Jiaotong University Affiliated Sixth People's Hospital, Shanghai, China. [6] College of Biomedical Engineering and Instrument Sciences, Zhejiang University, Hangzhou, China. [7] Department of Radiology, Huashan Hospital, Fudan University, Shanghai, China. [8] Department of Rehabilitation, Huashan Hospital, Fudan University, Shanghai, China. [9] MRC Cognition and Brain Sciences Unit, University of Cambridge, Cambridge, UK. [10] These authors contributed equally: Junhua Ding, Keliang Chen. [11] These authors jointly supervised: Qihao Guo, Zaizhu Han, Matthew. A. Lambon Ralph. ✉email: qhguo@sjtu.edu.cn; zzhhan@bnu.edu.cn; matt.lambon-ralph@mrc-cbu.cam.ac.uk

Recent years have shown a considerable increase of interest in the cognitive and behavioural functions of the anterior temporal lobes (ATL). This heightened attention on the ATL is founded upon detailed neuropsychological investigations of semantic dementia (SD)[1–5]. These have, in turn, inspired explorations of the ATL contribution to semantic representation in healthy participants[6–10], comparative studies across different patient groups with impaired semantic performance[11–16] and formal neuroanatomically constrained computational models of semantic cognition[17–21]. Accordingly, this study had integrated cognitive and clinical aims, not only to understand how semantic and other cognitive functions are distributed across the bilateral ATL and their sub-regions, but also how to conceive of the graded phenotype variations across SD. These dual aims can only be met by moving beyond investigating single cognitive activities in specific sub-groups of SD patients towards multi-task data collected in cohorts of SD patients. Such data allow us to elucidate the associations of different cognitive computations to one or more ATL sub-regions and also to derive a complete diagnostic cognitive-behavioural template such that all 'sub-types' of SD can be accurately diagnosed[22]. Without a picture of the entire disease, it is possible to miss patients or falsely sub-group patients. Indeed, there is a danger that graded variations in the same single disease entity are treated as multiple separate disease groups (e.g., the right temporal-variant of frontotemporal dementia (FTD), the semantic-variant of FTD, left temporal SD, etc[23]). This investigation went beyond these descriptions to generate a unified neurocognitive model of ATL functions and SD characteristics, which captures the known cognitive-behavioural variations across SD, maps these to the underlying patterns of atrophy, and integrates the considerable database on the graded bilateral ATL contributions to healthy semantic function[24] and semantic impairment in other patient groups[25]. The approach accommodates the facts that (i) there are not mutually exclusive sub-types of SD but rather gradely varying patterns of cognitive-behavioural presentation; (ii) SD is a part of the FTD spectrum; (iii) the cognitive-behavioural deficits reflect not only the balance of left vs. right ATL atrophy but also the total temporal lobe atrophy and its extension to insular and orbitofrontal cortex (OFC); and (iv) semantic representation in healthy participants appears to be supported by ATL regions, bilaterally, with graded variations in function reflecting differential patterns of connectivity.

The existing literature on SD and the associated explorations of ATL function, can be clustered around three descriptive 'sub-types'. The principal description is of the cognitively selective yet generalised conceptual degradation found in SD[1–3]. There is clear evidence that SD affects all categories of concept[26–28] in both receptive and expressive verbal and nonverbal tasks[29–34], with performance strongly influenced by concept familiarity, typicality and specificity[1,35,36]. This pervasive multimodal degradation led Snowden and Neary to coin the term "semantic dementia" in 1989[2]. Recent consensus criteria[37] proposed the alternative term "semantic-variant primary progressive aphasia (PPA)" thereby contrasting it from other PPAs. Although the patients' pronounced anomia and verbal comprehension deficit are often prominent initial clinical features[5,38], careful evaluation invariably identifies nonverbal comprehension deficits[31–34,39] even in early cases[30]. This striking clinical presentation is underpinned by bilateral atrophy centred on the polar and ventral aspects of the ATL[40,41]. Patients with more left than right (L > R) ATL atrophy are typically more common in most clinics[42]. Mild patients can present with predominantly left ATL atrophy or, more rarely, with right ATL atrophy; however, the disease is inherently bilateral in nature such that cases with asymmetric atrophy have considerable FDG-PET hypometabolism and rapid subsequent atrophy in the contralateral ATL[43,44]. The combination of multimodal semantic impairment with bilateral ATL atrophy is consistent with the hypothesis that semantic memory is underpinned by a bilaterally distributed representational system[20,25]. This notion is also supported by (i) patients with unilateral ATL damage (who have mild semantic impairment after left or right resection)[15]; (ii) bilateral ATL resection in non-human primates[45,46] and a single-case human study[47]—in which initial unilateral resection generated a mild multimodal semantic impairment and then subsequent contralateral ATL resection led to a profound deficit; (iii) results from ATL repetitive transcranial magnetic stimulation (rTMS) in healthy participants (which generates a transient, selective multimodal semantic deficit after left or right ATL stimulation)[48] and, (iv) fMRI investigations of semantic processing in healthy participants using methods that correct or minimise ATL-related distortion artefacts and other methodological limitations (which show bilateral ATL engagement by semantic tasks particularly in ventral and lateral regions)[8,24,49].

A second SD sample of patients presents early with predominantly left ATL atrophy, profound anomia accompanied by a mild comprehension deficit[17,38]. Again this pattern has direct parallels in other patient groups and healthy participants: (i) patients with left ATL damage/resection are significantly more anomic than their right counterparts[15,50–52]; (ii) fMRI studies involving speech production observe greater left than right ATL activations[24]; and (iii) rTMS has a larger effect on picture naming after left than right ATL stimulation[53]. Two hypotheses have been proposed to explain these data: (i) the left ATL houses lexical representations that support semantically driven speech production[50,51]; or (ii) that the bilateral ATL-hub semantic system connects to left-lateralised prefrontal speech production systems from the left ATL[17,20]. Although both theories explain the differential anomia of left ATL patients, the second approach is able to accommodate a range of additional findings, including some of those already noted above: (i) there are graded rather than absolute differences between left and right ATL cases/function; (ii) right unilateral ATL patients have some degree of anomia; (iii) careful examination of early L > R SD cases reveals mild deficits in verbal and nonverbal comprehension; (iv) the left ATL is activated in healthy participants for the same types of verbal and nonverbal comprehension tasks, and are compromised after rTMS.

The third SD sample is patients with greater right ATL atrophy, the least common in most clinics though the literature contains some notable single-case studies and small case-series[54–60]. Studies have focussed on two phenomena. Many very early, predominantly right ATL cases show visual recognition deficits for familiar people, followed with progression to anomia, generalised multimodal person semantic deficits and ultimately the generalised semantic impairment associated with SD[55,57]. A second literature relates to the social and behavioural impairments, which have been associated more with the R > L patients[42,60–63]. An important question to resolve is how face recognition and social-behavioural impairments fit with the other aspects of SD. Indeed, reaching a correct interpretation and unified model of SD is challenged by three facts: (i) the L > R patients also have behavioural changes when this is formally assessed[62] as well as poor semantic knowledge about people (and all other specific-level concepts)[1,18,36]; (ii) healthy participants activate superior and ventral ATL regions, bilaterally, when making social concept judgements[64–66] and exhibit transient impairments when the left or right ATL is stimulated[67–69]; and (iii) R > L patients tend to have more atrophy not only in bilateral temporal regions but also extending to OFC[62,63,70,71], which is known to be a behavioural-related region[72,73]. Even one seminal study that matched pairs of left- and right-dominant

cases for overall temporal atrophy found remaining differences in frontal atrophy[22].

The aim of the current study was to generate a data-driven, joint cognitive-neuroanatomical framework which assimilates these three clinical variations into a unified model and to test if the results align with those from contrastive patients (particularly those with unilateral ATL damage) and healthy participants. A recent computationally informed theory suggests that semantic representations are supported by a bilateral ATL system which engages multiple, distributed sources of verbal and sensory engrams to form coherent, generalisable concepts[25,52]. Graded variations in function both within (e.g., dorsal vs. ventral) and between (left vs. right) the ATL emerge as a natural consequence of differential connectivity to input and output systems[17,20,24,74,75]. For example, the heavy engagement of the left ATL in speech production[15,51,53,76] follows from connectivity to the strongly left-lateralised speech production systems[77].

The unified model is generated in two steps. The first uses multiple regression to explore the relationship between the primary SD symptoms and the integrity of frontotemporal regions. Multiple regression allows for shared variance in the distribution of atrophy and symptoms across patients to be modelled and separated. In the second step we utilise a data-driven approach (principal component analysis (PCA) with varimax rotation) to explore the underlying, graded dimensions in the patients' neuropsychological and social-behavioural data. This approach dispenses mutually exclusive categories (which may not exist) and, instead, assumes that the patients vary in graded ways across one or more underlying dimensions[72,78,79]. Then the factor scores (rather than individual tests) are related to the patients' atrophy (univariate and multivariate whole-brain analyses). This PCA plus voxel-symptom mapping has been used by multiple independent research groups with post-stroke aphasia data, leading to consistent patterns of symptom dimensions and neural correlates[78–83] as well as one recent exploration of impulsivity and apathy in frontotemporal lobar syndromes[72]. We extend the same methodology to a large SD sample containing variation of left-right ATL atrophy and disease severity.

## Results

**The neuropsychological profiles of SD patients**. Table 1 summarises the SD patients' demographic, neuropsychological and behavioural information. Patients were matched to the healthy controls in terms of age, sex and education years ($p$-values > 0.07).

**Table 1 Task performance and cerebral volumes of SD patients and healthy subjects.**

| | Healthy controls | SD patients | |
| --- | --- | --- | --- |
| | (n = 20) | left SD (n = 28) | right SD (n = 19) |
| Demographic characteristics | | | |
| Age (years) | 61 (4)[0.26] | 62 (7)[0.71] | 63 (7)[0.17] |
| Sex (male: female) | 8:12[0.24] | 16:12[0.51] | 9:10[0.64] |
| Education level (years) | 10 (3)[0.07] | 12 (3)[0.16] | 11 (3)[0.73] |
| Disease duration (years) | | 3 (2)[0.50] | 3 (2) |
| Task performance | | | |
| Object semantics | | | |
| Oral picture naming | 92% (4%)[3*10ˆ−17] | 30% (19%)[0.007] | 46% (20%)[6*10ˆ−9] |
| Word-picture verification | 96% (3%)[10ˆ−7] | 67% (21%)[0.47] | 71% (20%)[0.00003] |
| Picture associative matching | 95% (4%)[2*10ˆ−8] | 78% (11%)[0.67] | 77% (8%)[3*10ˆ−9] |
| Face semantics | | | |
| Oral picture naming | 78% (16%)[5*10ˆ−16] | 10% (9%)[0.29] | 13% (13%)[5*10ˆ−16] |
| Word-picture verification | 93% (9%)[4*10ˆ−10] | 54% (22%)[0.03] | 38% (22%)[8*10ˆ−10] |
| Picture associative matching | 96% (6%)[2*10ˆ−8] | 72% (17%)[0.54] | 69% (14%)[3*10ˆ−8] |
| Visual perception | | | |
| Object perception | 96% (5%)[0.000003] | 72% (20%)[0.005] | 87% (13%)[0.01] |
| Face perception | 78% (14%)[0.0001] | 58% (18%)[0.48] | 54% (18%)[0.00004] |
| General cognition | | | |
| MMSE | 28 (1)[9*10ˆ−9] | 22 (4)[0.35] | 23 (4)[0.00001] |
| NPI (0:1:2:3) | | | |
| Agitation | | 7:4:1:1[0.70] | 5:5:3:1 |
| Depression | | 8:2:2:1[0.35] | 5:5:1:3 |
| Anxiety | | 6:3:4:0[0.09] | 6:7:0:1 |
| Apathy | | 5:5:3:0[0.62] | 3:7:3:1 |
| Disinhibition | | 9:2:1:1[0.46] | 6:6:1:1 |
| Irritability | | 6:4:3:0[0.56] | 5:4:3:2 |
| Cerebral volume | | | |
| Total grey matter volume (mm³) | 32,156 (2496)[0.006] | 29,522 (3508)[0.26] | 28,319 (3624)[0.001] |
| Left − right ATL volume (mm³) | −10 (43)[3*10ˆ−15] | −191 (65)[7*10ˆ−21] | 167 (82)[6*10ˆ−9] |
| Left + right ATL volume (mm³) | 1625 (163)[5*10ˆ−12] | 1092 (238)[0.01] | 942 (148)[5*10ˆ−16] |
| Left + right OFC volume (mm³) | 1144 (94)[0.00008] | 1002 (122)[0.01] | 897 (155)[0.000002] |
| Left + right hippocampus volume (mm³) | 487 (34)[4*10ˆ−10] | 386 (54)[0.25] | 369 (42)[10ˆ−11] |

Sex and NPI scores were compared between each two groups by $\chi^2$ tests (df = 1 and 3, respectively). Other scores were compared by two-tailed independent $t$-tests. The equality of variances for $t$-tests was tested by the Levene's test. If the assumption of equality of variances was rejected, the df would be corrected correspondingly. The numbers in parentheses are the standard deviations. The superscripts of controls indicate the uncorrected $p$-values of comparisons between left SD and controls (df for $t$-tests = 46, except for word-picture verification and visual perception, whose df = 45 and 44, respectively). The superscripts of left SD indicate the uncorrected $p$-values of comparisons between left SD and right SD (df for $t$-tests = 37, except for word-picture verification and visual perception, whose df = 36 and 34, respectively). The superscripts of right SD indicate the uncorrected $p$-values of comparisons between right SD and controls (df for $t$-tests = 45, except for visual perception, whose df = 44). Source data are provided as a Source data file.
*SD* semantic dementia, *MMSE* Mini Mental State Examination, *NPI* neuropsychiatric inventory, *ATL* anterior temporal lobe, *OFC* orbital frontal cortex

Both left and right SD patients presented with impairments of object and face semantics, visual perception and general cognitive ability (p-values < 0.01). In addition, some behavioural problems were reported. The most six frequent symptoms included apathy (70%), irritability (59%), agitation (56%), anxiety (56%), depression (52%), and disinhibition (44%). With regard to cognitive and behavioural differences between left and right SD patients, left SD patients performed better on word-picture verification of faces (mean difference = 15% confidence interval (CI) = 2–29%, $t(44) = 2.32$, $p = 0.03$), while right SD patients had better

performance on visual object perception (mean difference = −15%, 95% CI = −25 to −5%, $t(42) = −3.00$, $p = 0.005$) and picture naming of objects (mean difference = 16%, 95% CI = −28 to −5%, $t(45) = −2.82$, $p = 0.007$).

**The atrophy of SD patients.** The distribution of atrophy in this SD sample was typical of that reported by other research groups. Specifically, the voxel-based analysis revealed that both left and right SD patients had marked atrophy in the bilateral temporal lobes, insula and OFC (Fig. 1a, b; FDR-corrected $p < 0.05$). When

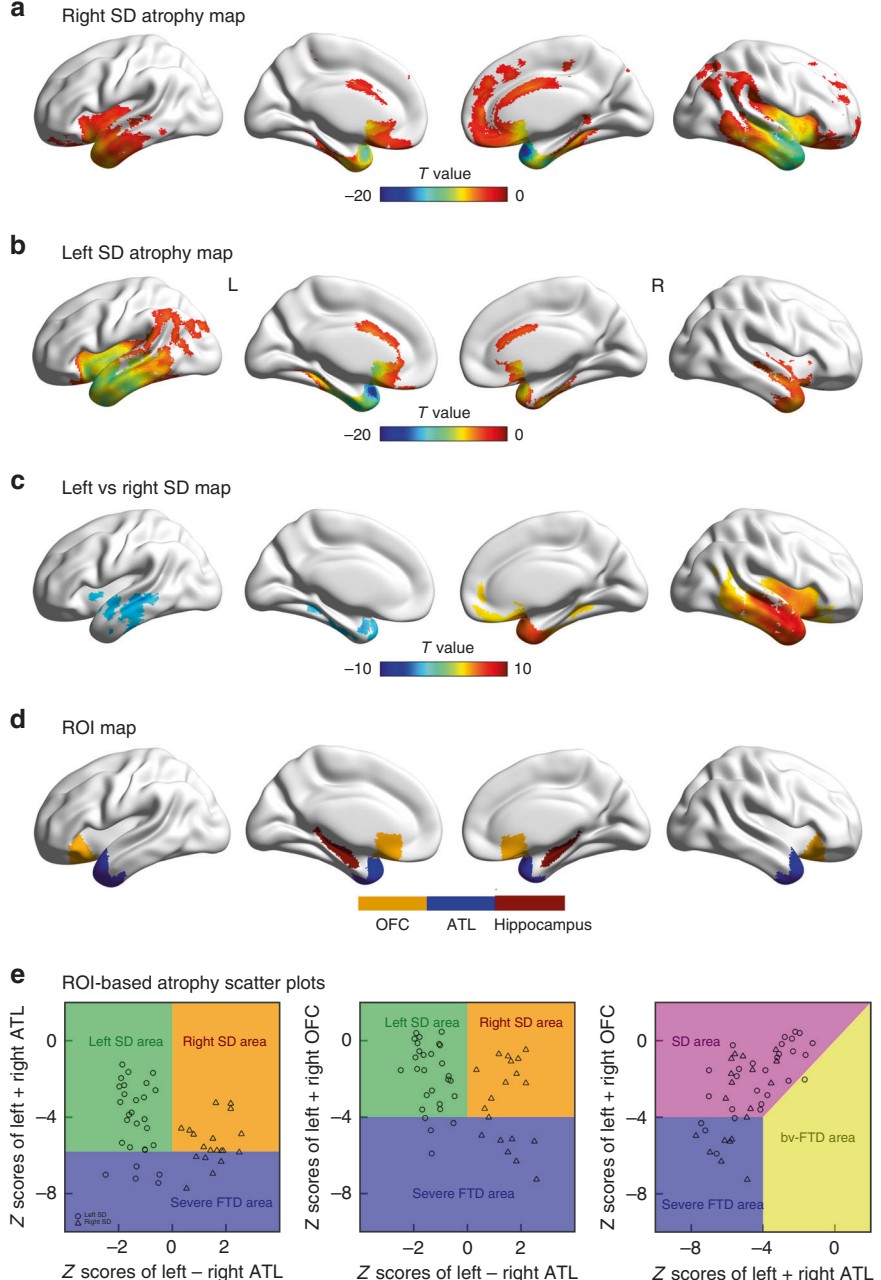

**Fig. 1 The atrophy pattern of 47 SD patients. a**, **b** The comparisons between left SD ($n = 28$) vs. controls ($n = 20$) and right SD ($n = 19$) vs. controls. The two-sided voxel-based independent $t$-tests were used (df = 46 and 37, respectively; FDR-corrected $p < 0.05$). **c** the comparison between left and right SD. The two-sided voxel-based independent $t$-test was used (df = 45; FDR-corrected $p < 0.05$). **d** the ROIs used in the multiple regression analyses. **e** Scatterplots showing the distribution and inter-relationships of left ATL, right ATL and OFC atrophy across the left and right-sided SD patients and the FTD sub-categories. The boundaries of severe FTD and SD/bv-FTD were defined in terms of containing 25% of the cases with the most severe bilateral ATL and OFC atrophy, respectively. ATL anterior temporal lobe, OFC orbital frontal cortex, SD semantic dementia, bv-FTD behavioural-variant frontotemporal dementia. Source data are provided as a Source Data file.

directly comparing the two SD sub-groups, right SD showed more atrophy in the right temporal lobe, insula and OFC, whereas additional atrophy in the left SD sub-group was mainly restricted to the left temporal lobe (Fig. 1c; FDR-corrected $p < 0.05$).

Simple comparison of left and right SD patients is complicated by the fact that other potentially important differences occur across the groups. In keeping with previous reports[62,70], the ROI analysis confirmed: (i) atrophy for both left and right SD patient groups was found in bilateral ATL and OFC relative to controls ($p$-values $< 0.00008$; Table 1); and (ii) comparison between left and right SD revealed that the right SD patients had more atrophy to both ATL (mean difference $= 151$ mm$^3$, 95% CI $= 37$–264 mm$^3$, $t(45) = 2.67$, $p = 0.01$) and OFC (mean difference $= 105$ mm$^3$, 95% CI $= 23$–186 mm$^3$, $t(45) = 2.59$; $p = 0.01$; Table 1). This is important for interpretation of any cognitive or behavioural differences between left and right SD patients for two reasons: (i) the right SD patients were neuroanatomically more severe and (ii) their atrophy encroached into potentially important regions such as the left ATL and OFC. These relationships can be clearly observed in the scatterplots (Fig. 1e): (i) there is no boundary between left and right SD cases; (ii) most patients have a degree of bilateral ATL and OFC damage, with the exception of some very mild left SD cases (indeed, OFC damage is correlated with total temporal damage ($r(45) = 0.64$, 95% CI $= 0.43$–0.78, $p = 0.000001$; Fig. 1e, right panel; though, as expected for an SD sample, there is always greater temporal than OFC atrophy; replicating the results from Seeley et al.[22]); (iii) the right SD cases have greater bilateral ATL (Fig. 1e, left panel) and OFC (Fig. 1e, middle panel) damage; and (iv) very mild right ATL-only cases are extremely rare (indeed, like many SD group studies, no such cases were found in our clinical sample; thus there are no

cases in the upper right-hand region of the left scatterplot in Fig. 1e). Accordingly, rather than using a simple left-right binary division of SD cases, we adopted statistical approaches which take into account not only the laterality but also total temporal and OFC atrophy. We first used simultaneous regression (to map these anatomical measures to the cognitive and behavioural results) and then we utilised PCA to establish the underlying dimensions of cognitive-behavioural variation in the SD sample and explored their relationship to the atrophy distribution.

**The relation between atrophy and task performance.** Simultaneous regression analyses were conducted using the four ROI atrophy measures to predict each task. As shown in Table 2, for object word-picture verification, the sum of ATL volume was the only significant variable ($beta = 0.44$, 95% CI $= 0.06$–0.82, $t(45) = 2.30$, $p = 0.03$). Two picture naming tasks were predicted by the sum of ATL volume (object: $beta = 0.39$, 95% CI $= 0.05$–0.72, $t(46) = 2.28$, $p$-values $= 0.03$; face: $beta = 0.42$, 95% CI $= 0.04$–0.81, $t(46) = 2.19$, $p = 0.03$) plus the difference of left > right atrophy (object: $beta = 0.53$, 95% CI $= 0.26$–0.79, $t(46) = 3.95$, $p$-values $= 0.0003$; face: $beta = 0.35$, 95% CI $= 0.04$–0.65, $t(46) = 2.27$, $p = 0.03$). Regarding the NPI measures, only anxiety and apathy showed significant relationships with atrophy (note: agitation, depression, disinhibition and irritability were rated as 0 or 1 for most SD participants and thus there is a ceiling effect for these behavioural measures). Anxiety levels exhibited a significant negative relationship ($beta = -0.53$, 95% CI $= -1.04$ to $-0.02$, $t(26) = -2.09$, $p = 0.05$) with total hippocampus atrophy (perhaps reflecting heightened anxiety with the onset and diagnosis of the disease). Apathy was positively

**Table 2** *Beta* values of four ROI variables predicting task and factor scores.

| | Left − right ATL volume | Left + right ATL volume | Left + right OFC volume | Left + right hippocampus volume |
|---|---|---|---|---|
| **Face semantics** | | | | |
| Oral picture naming | 0.35 (0.04-0.65)$^{0.03}$ | 0.42 (0.04-0.81)$^{0.03}$ | −0.06 (−0.50-0.39)$^{0.81}$ | 0.06 (−0.38-0.49)$^{0.80}$ |
| Word-picture verification | −0.11 (−0.40-0.19)$^{0.47}$ | 0.34 (−0.04-0.71)$^{0.08}$ | −0.12 (−0.56-0.69)$^{0.60}$ | 0.27 (−0.16-0.69)$^{0.22}$ |
| Picture associative matching | 0.07 (−0.23-0.37)$^{0.64}$ | 0.26 (−0.12-0.64)$^{0.19}$ | −0.01 (−0.46-0.43)$^{0.95}$ | 0.29 (−0.14-0.72)$^{0.19}$ |
| **Object semantics** | | | | |
| Oral picture naming | 0.53 (0.26-0.79)$^{0.0003}$ | 0.39 (0.05-0.72)$^{0.03}$ | −0.36 (−0.75-0.03)$^{0.07}$ | 0.33 (−0.05-0.71)$^{0.10}$ |
| Word-picture verification | 0.27 (−0.03-0.57)$^{0.08}$ | 0.44 (0.06-0.82)$^{0.03}$ | −0.30 (−0.74-0.14)$^{0.18}$ | 0.28 (−0.15-0.71)$^{0.20}$ |
| Picture associative matching | −0.09 (−0.40-0.22)$^{0.56}$ | 0.10 (−0.30-0.49)$^{0.63}$ | −0.07 (−0.53-0.38)$^{0.76}$ | 0.36 (−0.09-0.80)$^{0.12}$ |
| **Visual perception** | | | | |
| Object perception | 0.34 (0.00-0.69)$^{0.05}$ | −0.07 (−0.50-0.36)$^{0.75}$ | 0.16 (−0.34-0.66)$^{0.52}$ | 0.03 (−0.45-0.52)$^{0.89}$ |
| Face perception | −0.11 (−0.43-0.20)$^{0.47}$ | −0.16 (−0.56-0.23)$^{0.41}$ | 0.34 (−0.12-0.79)$^{0.15}$ | 0.24 (−0.20-0.69)$^{0.28}$ |
| **NPI** | | | | |
| Agitation | −0.17 (−0.60-0.26)$^{0.43}$ | −0.29 (−0.84-0.26)$^{0.30}$ | 0.14 (−0.43-0.70)$^{0.63}$ | 0.31 (−0.30-0.92)$^{0.32}$ |
| Depression | −0.12 (−0.56-0.31)$^{0.57}$ | −0.2 (−0.76-0.35)$^{0.47}$ | −0.01 (−0.58-0.56)$^{0.97}$ | −0.15 (−0.77-0.46)$^{0.62}$ |
| Anxiety | 0.13 (−0.23-0.49)$^{0.48}$ | −0.26 (−0.72-0.21)$^{0.27}$ | 0.24 (−0.23-0.71)$^{0.31}$ | −0.53 (−1.04-0.02)$^{0.05}$ |
| Apathy | −0.01 (−0.38-0.35)$^{0.95}$ | −0.28 (−0.75-0.19)$^{0.24}$ | 0.79 (0.31-1.26)$^{0.003}$ | −0.45 (−0.96-0.07)$^{0.09}$ |
| Disinhibition | −0.25 (−0.68-0.18)$^{0.25}$ | −0.14 (−0.69-0.41)$^{0.61}$ | 0.09 (−0.46-0.65)$^{0.73}$ | −0.24 (−0.85-0.36)$^{0.42}$ |
| Irritability | −0.05 (−0.48-0.38)$^{0.82}$ | 0.06 (−0.49-0.62)$^{0.82}$ | 0.16 (−0.41-0.72)$^{0.58}$ | −0.45 (−1.06-0.16)$^{0.15}$ |
| **PCA** | | | | |
| Factor 1: apathy | 0.23 (−0.09-0.55)$^{0.17}$ | 0.03 (−0.38-0.44)$^{0.88}$ | 0.39 (−0.08-0.87)$^{0.11}$ | −0.37 (−0.84-0.09)$^{0.12}$ |
| Factor 2: face | −0.19 (−0.46-0.08)$^{0.16}$ | 0.12 (−0.22-0.46)$^{0.49}$ | 0.16 (−0.23-0.56)$^{0.42}$ | 0.29 (−0.10-0.68)$^{0.15}$ |
| Factor 3: naming | 0.61 (0.37-0.86)$^{0.00001}$ | 0.49 (0.18-0.81)$^{0.003}$ | −0.34 (−0.70-0.03)$^{0.07}$ | 0.10 (−0.26-0.46)$^{0.57}$ |
| Factor 4: disinhibition | −0.13 (−0.47-0.20)$^{0.44}$ | −0.02 (−0.45-0.40)$^{0.91}$ | 0.11 (−0.38-0.61)$^{0.65}$ | −0.04 (−0.52-0.44)$^{0.87}$ |
| Factor 5: SD severity | 0.17 (−0.15-0.48)$^{0.30}$ | −0.22 (−0.62-0.18)$^{0.28}$ | 0.4 (−0.06-0.86)$^{0.09}$ | 0.13 (−0.32-0.59)$^{0.56}$ |

The task and factor scores were predicted by multiple linear regressions ($n = 46$, 44, 27 and 47 for word-picture verification, visual perception, NPI and others). The significance of *beta* values was tested by one sample *t*-tests compared with 0 (df = 45, 43, 26 and 46 for word-picture verification, visual perception, NPI and others). The numbers in superscripts are the uncorrected *p*-values. The numbers in parentheses are the 95% confidence intervals. Source data are provided as a Source data file.
*ATL* anterior temporal lobe, *OFC* orbital frontal cortex, *NPI* neuropsychiatric inventory, *PCA* principal component analysis.

related to OFC atrophy (beta = 0.79, 95% CI = 0.31–1.26, t(26) = 3.36, p = 0.003).

**PCA of task performance**. All neuropsychological and behavioural data were entered into a PCA. The KMO value for the resultant model was 0.71. Five factors were found as the optimal solution for our data, and then the missing data were imputed by probabilistic PCA. Using the imputed data, PCA was performed with varimax rotation. Figure 2 (top panels) displays the loadings of the tasks on each of the orthogonal factors. Five factors with an eigenvalue >1 were extracted, which accounted for 85% of the total variance. Factor 1 accounted for 21% of the variance, with high loadings of apathy (0.74), depression (0.89) and anxiety (0.72), and thus we refer to this factor as 'apathy'. Factor 2 accounted for 20% of the variance. High loadings were found on the visual aspects of face recognition (rather than semantics more generally), specifically face matching (0.84) and face verification (0.84); thus we refer to this as the 'face' factor. Factor 3 (variance = 16%) was interpreted as 'naming' because it heavily loaded on three verbal tasks (face naming: 0.75; object naming: 0.90; object verification: 0.69). Factor 4 (variance = 16%) was labelled as 'disinhibition' due to its high loadings with disinhibition (0.89), agitation (0.88) and irritability (0.65). Factor 5 accounted for 12% of variance, which had high positive loadings on object (0.86) and face (0.77) perception. Because perceptual impairments only occur in the very late stage of SD, we refer to this factor as 'SD severity'.

**The relation between atrophy and PCA factors**. In the first symptom-atrophy mapping analysis, we used the four ROI variables in a multiple regression to predict the five PCA scores (as done for the individual test scores, see above). The sum and difference of bilateral ATL volumes showed significant effects for the 'naming' factor (ATL sum: beta = 0.49, 95% CI = 0.18–0.81, t(46) = 3.12, p = 0.003; ATL difference: beta = 0.61, 95% CI = 0.37–0.86, t(46) = 4.92, p = 0.00001; Table 2). None of the ROI variables were significantly related to the 'disinhibition', 'apathy', 'SD severity' and 'face' factors.

Rather than limiting the symptom-atrophy mapping to the ROI regions, we also utilised voxel-based correlation mapping (VBCM) to provide a whole-brain analysis. These results replicated those found in the ROI-based analysis and revealed additional areas of interest (lower panels in Fig. 2 and Supplementary Tables 1–3). The univariate and multivariate VBCM obtained similar results. The 'SD severity' factor was correlated with bilateral middle cingulate gyri, posterior temporal and parietal regions; i.e., the edges of the atrophy distribution in SD (Fig. 1a, b) and the progression of atrophy observed in longitudinal studies[44,84,85]. The 'naming' factor was positively related to the atrophy of the left ATL. The right dorsal superior temporal gyrus (STG) and rectus gyrus were the only regions related to the 'inhibition' factor. The 'apathy' factor was associated with the bilateral medial frontal cortices only. Finally, the 'face-recognition' factor was related to various areas within the right temporal lobe, insula and bilateral medial frontal lobes.

**Analysis of face-related ROIs in SD**. Given that the PCA-VBCM highlighted various areas beyond the right ATL, we explored the relationship of atrophy in five face-related ROIs (derived from studies of healthy participants) and the face-related PCA factor. Atrophy in all five face-related ROIs was highly correlated with the 'face' factor (r(45) values > 0.37, p-values < 0.01; Fig. 3). Further validation analyses are reported in the Supplementary Tables 4–6, which revealed that this result for face ROIs was specific to face but not object processing, specific to SD patients

but not normal controls, and only to right-sided regions but not the corresponding left homologues.

**Extrapolation to early predominantly right-sided SD**. The preceding analyses indicate that SD patients can be conceptualised within a unified multidimensional model. If correct then it should be possible not only to interpolate within the existing data but, more challengingly, to extrapolate to regions of the space for which there were no data. Like many other SD group samples, there were no patients with very mild, predominantly right ATL atrophy. Extrapolating from the existing model, the presentation of very early right ATL-only patients should be dominated by the right ATL loading factors 2 and 4, namely visual face recognition and positive behavioural deficits. Then, longitudinally, not only would there be augmented atrophy in the right ATL but it should also become more bilateral in nature. Ultimately, atrophy should also encroach on the insula and OFC. Accordingly, the initial face recognition and positive behavioural deficits should be joined next by generalised semantic and naming impairments (associated with bilateral ATL atrophy) and then with negative behavioural features (associated with the OFC atrophy).

To test this hypothesis, we conducted a systematic review of the literature, selecting longitudinal case reports of patients with early right-predominant ATL atrophy. In particular, we focussed on investigations that provided detailed neuropsychological evaluation of the factors of interest (face recognition, naming, semantic abilities and behavioural impairment) as well as information about how these progressed over time. We identified 15 cases in the literature and a summary of their symptom progression is provided in Table 3. With one exception, all other cases reflected the predicted neuropsychological pattern; namely, patients presented first with early visual face recognition or positive behavioural deficits followed by more general semantic, naming and negative behavioural problems.

**Discussion**

The purpose of this investigation was to generate a unified model of ATL functions, which captures the known cognitive-behavioural variations in SD, maps these to the underlying patterns of atrophy, and integrates with the considerable database on healthy semantic function and semantic impairment in other patient groups (for a recent review, see Lambon Ralph et al.[25]). This aim was achieved through the analysis of a large SD case-series that varied in terms of severity and the balance of left vs. right ATL atrophy (i.e., representing the typical distribution of cases[42]). Rather than forcing a categorical framework onto the continuous variations across the patients, our analytical approach was able to capture the graded neuropsychological differences and map these to the patients' distribution of frontotemporal atrophy. As expected[2], the multiple regression analyses confirmed that the degree of generalised verbal and nonverbal semantic impairment was related to the patients' total, bilateral ATL atrophy. Verbal production and word-finding abilities were related to total ATL atrophy as well as to the balance of left > right ATL atrophy. Behavioural apathy was found to relate positively to the degree of orbitofrontal atrophy. The data-driven PCA replicated and extended these findings by identifying five statistically independent cognitive-behavioural factors and their unique atrophy correlates. A generalised severity factor was related to increased atrophy around the perimeter of the frontotemporal regions implicated in SD. Again, naming was uniquely correlated with the degree of left ATL atrophy and apathy to medial OFC volumes. In addition, disinhibited behaviour was uniquely correlated with

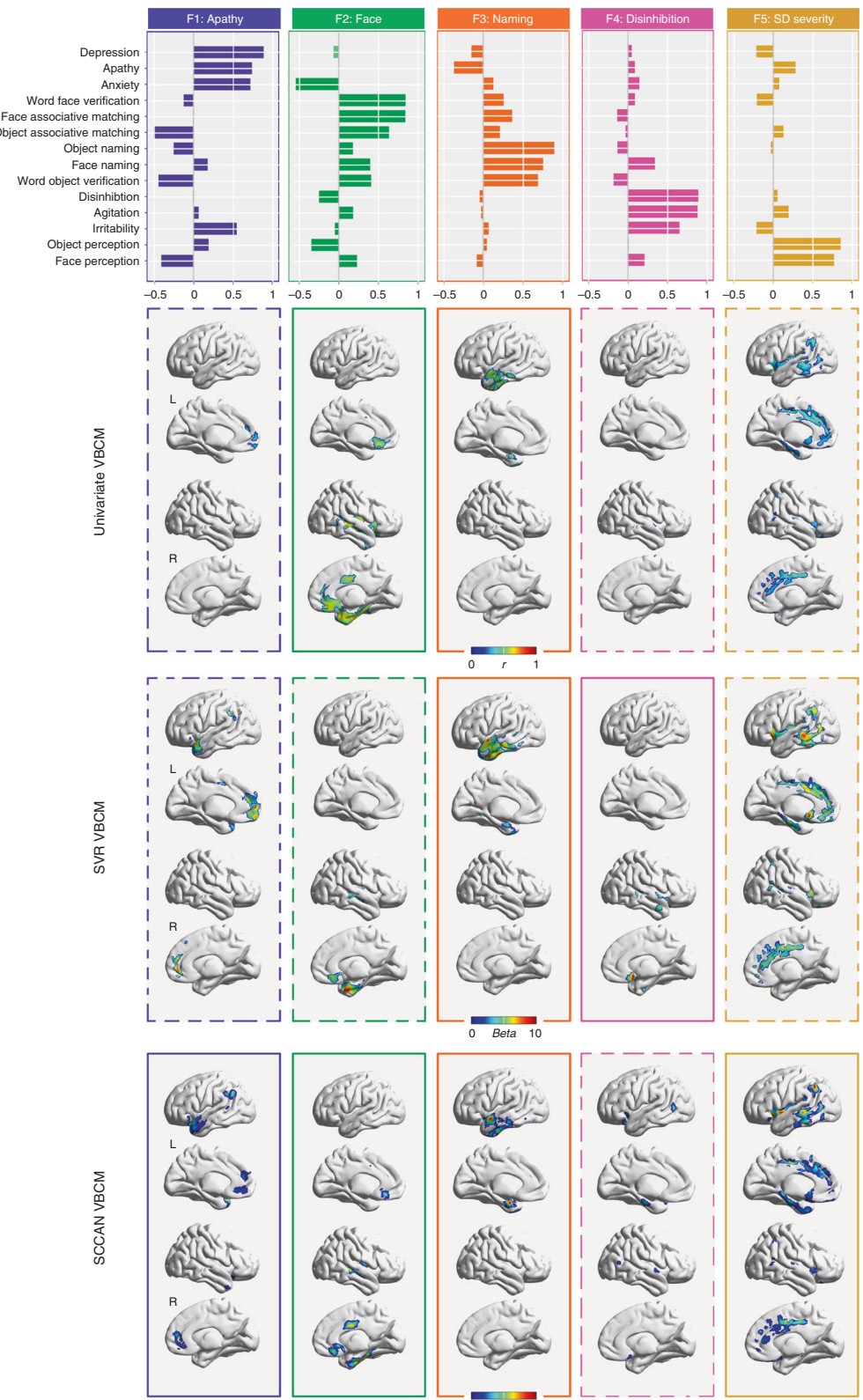

**Fig. 2 The factor loadings of PCA and corresponding maps of the significant regions associated with each factor (cluster size > 50 voxels; $n = 47$).** Two-tailed VBCM analyses were used (df = 45). Here, only positive clusters are reported. The solid and dashed lines of univariate VBCM reflect the multiple comparison corrected and uncorrected results, respectively. The solid and dashed lines of multivariate VBCM reflect the results with cross-validation p-values < 0.08 and < 0.2, respectively (SVR: F1 $p = 0.18$; F2 $p = 0.17$; F3 $p = 0.002$; F4 $p = 0.04$; F5 $p = 0.17$; SCCAN: F1 $p = 0.08$; F2 $p = 0.00003$; F3 $p = 0.00002$; F4 $p = 0.17$; F5 $p = 0.04$). VBCM voxel-based correlation mapping, SVR support vector regression, SCCAN sparse canonical correlation analysis for neuroimaging. Source data are provided as a Source data file.

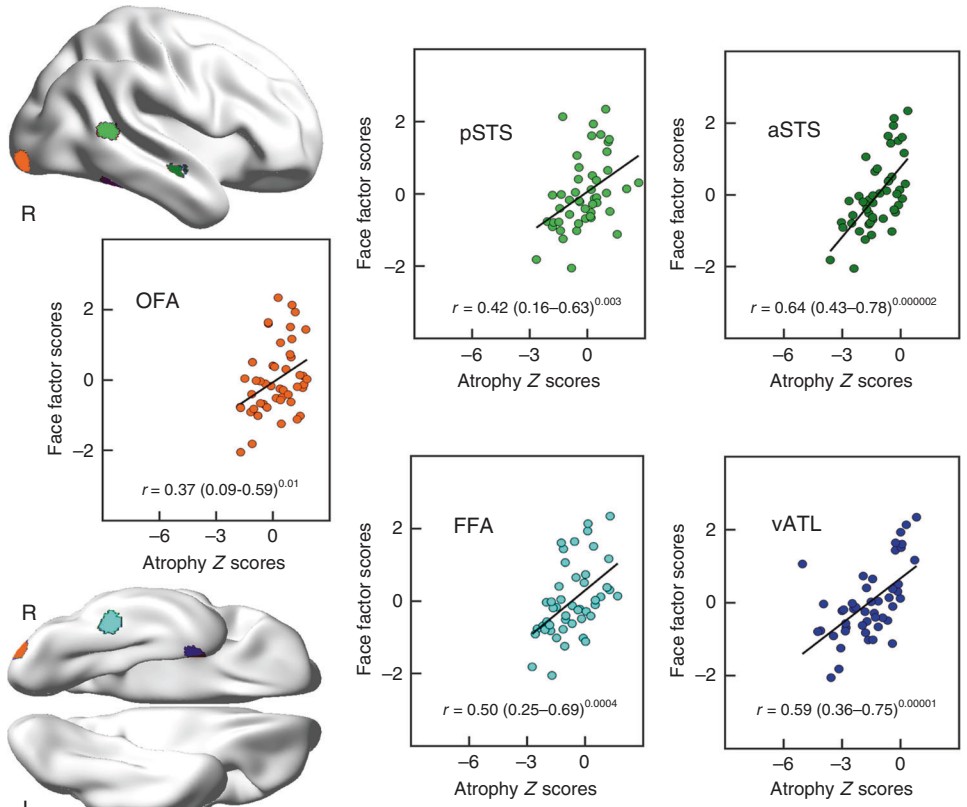

**Fig. 3 Two-tailed Pearson correlations between face-related ROIs' atrophy and 'face' factor scores ($n = 47$; df $= 45$).** The brain maps show the locations of ROIs. The numbers in parentheses are the 95% confidence intervals. The numbers in superscripts are the uncorrected p-values. OFA occipital face area, pSTS posterior superior temporal sulcus, aSTS anterior superior temporal sulcus, FFA fusiform face area, vATL ventral anterior temporal lobe. Source data are provided as a Source data file.

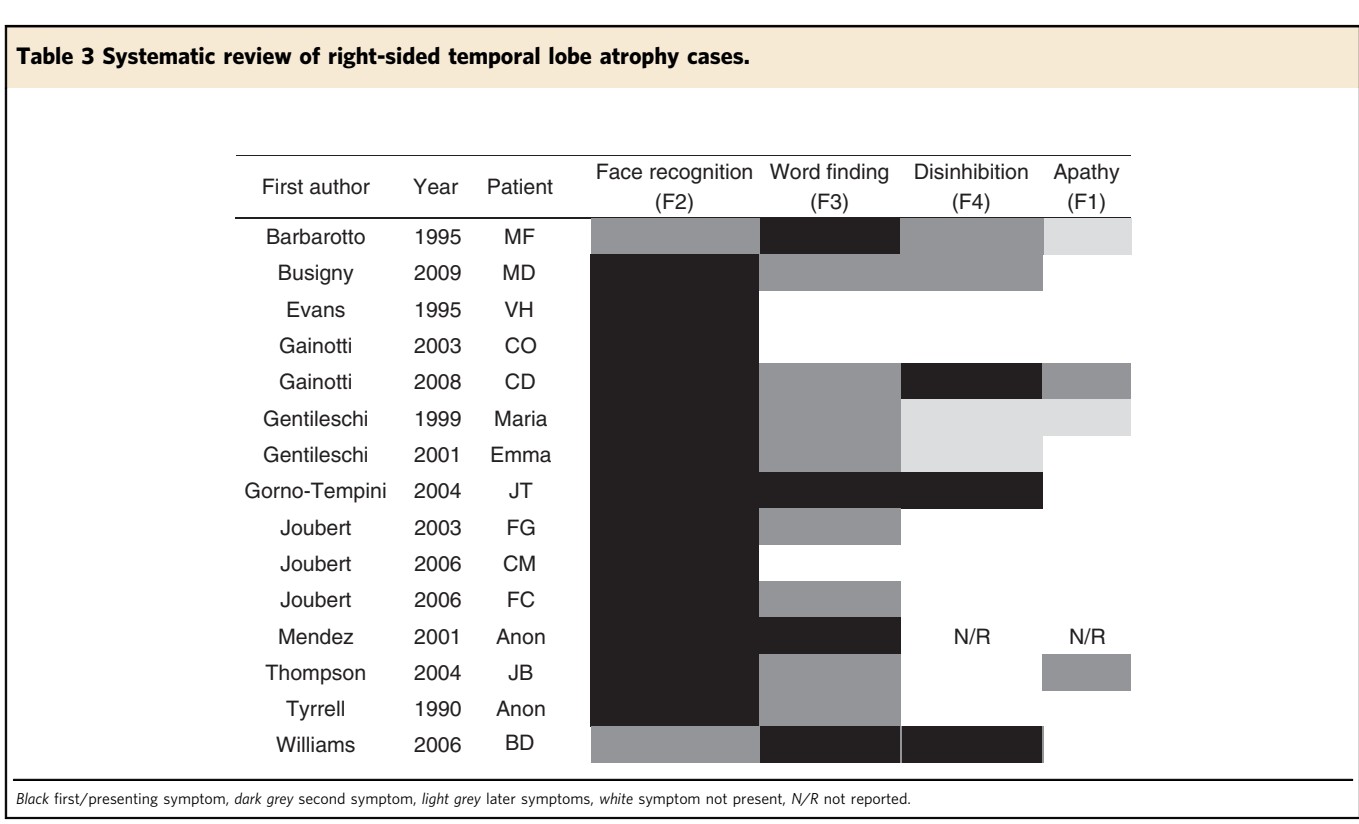

**Table 3 Systematic review of right-sided temporal lobe atrophy cases.**

| First author | Year | Patient | Face recognition (F2) | Word finding (F3) | Disinhibition (F4) | Apathy (F1) |
|---|---|---|---|---|---|---|
| Barbarotto | 1995 | MF | | | | |
| Busigny | 2009 | MD | | | | |
| Evans | 1995 | VH | | | | |
| Gainotti | 2003 | CO | | | | |
| Gainotti | 2008 | CD | | | | |
| Gentileschi | 1999 | Maria | | | | |
| Gentileschi | 2001 | Emma | | | | |
| Gorno-Tempini | 2004 | JT | | | | |
| Joubert | 2003 | FG | | | | |
| Joubert | 2006 | CM | | | | |
| Joubert | 2006 | FC | | | | |
| Mendez | 2001 | Anon | | | N/R | N/R |
| Thompson | 2004 | JB | | | | |
| Tyrrell | 1990 | Anon | | | | |
| Williams | 2006 | BD | | | | |

*Black* first/presenting symptom, *dark grey* second symptom, *light grey* later symptoms, *white* symptom not present, *N/R* not reported.

right dorsal STG and OFC atrophy and face recognition to right ATL volumes.

It is important to consider what these PCA dimensions mean. Although it might be tempting to think of each dimension as a clinical sub-group, this is not correct. By definition, PCA and other similar statistical techniques attempt to find continuous, graded dimensions underpinning the variations in the observed data, rather than identify clusters of cases. There is an example of this in the current study; although, like most clinical SD samples[42], there were no patients with early right ATL-only atrophy, the PCA was able to reveal that (i) there was independent variation in the degree of face-recognition deficits and (ii) in turn, this was related to the degree of right ATL atrophy. In effect, the multiple regression and PCA analyses offer a unified model of SD by identifying and quantifying the factors that underpin the patients' clinical variations (i.e., semantic impairment, anomia, prosopagnosia, disinhibited behaviour and apathy). In turn, when these factors are related to the distribution of atrophy, the resultant maps show regions that are uniquely related to each factor (areas that contribute to more than one function are not identified through this method but are identified by the regression analyses). Of course, it should be noted that all PCA are limited to the assessments included and additional dimensions might emerge when new types of assessment are added[79].

The paradigmatic symptom of SD is their selective yet progressive, multimodal semantic impairment[2,3]. This central symptom was found to relate to the degree of bilateral ATL atrophy. This finding aligns closely with data from healthy participants, other patient groups as well as comparative neurosurgical studies (for reviews, see Lambon Ralph et al.[25] and Rice et al.[75]). For example, distortion-corrected/avoiding fMRI identifies bilateral ATL activations when healthy participants complete various semantic tasks[8,66], rTMS to left or right ATL generates a transient, selective semantic impairment[7,48,86], and patients with unilateral ATL resection for temporal lobe epilepsy have a mild semantic impairment albeit much less pronounced than that observed in SD[15,87]. The difference between unilateral and bilateral damage was first demonstrated in non-human primates[45,46] and one human case[47]; initial unilateral resection generating a transient multimodal semantic impairment leading to a considerable, chronic deficit after bilateral removal. Thus, the two ATL work in concert to generate a robust semantic system which is only significantly compromised in bilateral diseases. This bilateral hypothesis is supported by formal computational models of a bilateral semantic system[20] as well as combined rTMS-fMRI explorations (which show that after left ATL rTMS in healthy participants, there is both upregulation of activity in the right ATL and increased inter-hemispheric functional connectivity)[66,88].

The second dimension of SD is anomia, which is probably the most common, presenting symptom. In keeping with previous explorations[17,89,90], the patients' anomia was found not only to be related to the degree of general semantic impairment (bilateral ATL volume) but also to the left > right ATL atrophy balance. This result aligns directly with convergent data from fMRI and rTMS in healthy participants[24,53] and from patients with unilateral ATL damage[15,51,87,91]—all pointing to a greater role of left than right ATL in semantically initiated speech production/naming. Formal computational models have shown how this form of asymmetric involvement in naming can arise from an inherently bilateral ATL semantic system. Specifically, differential connectivity to the left hemisphere prefrontal speech output systems means that the left ATL component becomes especially important in driving speech from semantic input[17,20,92].

The face-recognition-right ATL dimension may reflect a complementary effect of differential connectivity, this time with respect to input to the bilateral ATL. Semantic knowledge about people aligned with general semantic knowledge, and was associated with the degree of bilateral ATL damage. In contrast, the degree of right ATL atrophy was linked with visual face recognition per se (cf. the classical definition of visual prosopagnosia)[90,93]. This finding is consistent with patients with unilateral right ATL resection who show greater deficits of visual recognition of familiar people[87,94]. Furthermore, the progression of rare, early right ATL SD patients (Table 3) fits with this result; in the very earliest stage (which is long before most right > left SD patients present to clinic), right-only SD patients are reported to have visual prosopagnosia (i.e., poor recognition from faces with good semantic knowledge of the same people when probed from another input modality) which later develops into a generalised semantic impairment and anomia, presumably when bilateral ATL pathology has begun to evolve. The differential right > left ATL involvement in face recognition may again reflect differential connectivity[24,87]. It is well established that there is a strong asymmetry in ventral occipital-temporal regions for different visual objects[95], with face processing exhibiting a rightward bias[96] and the opposite for word recognition[97]. Extending the logic and computational demonstrations for the effect of differential connectivity on function[17,20,74], atrophy of right ATL regions might impact much earlier than left ATL atrophy on face recognition because there is stronger visual face input to this part of the bilaterally distributed semantic system. Atrophy might also extend posteriorly into the FFA and other parts of the right-lateralised extended face processing network[98–101]. In our further analysis, the face deficits of SD were found to be correlated with the level of atrophy in all of the nodes in this network.

The two remaining dimensions relate to the behavioural changes observed in SD and FTD patients more generally[73,102]. The level of disinhibited behaviour (i.e., disinhibition, irritability and agitation) related to both right dorsal STG and OFC volumes. This result aligns with findings from three other literatures (for the literature review of right dorsal STG, see Table 4 and Fig. 4): (i) previous studies have associated disinhibited behaviour with the right > left SD patients[60–62,103], entire FTD and behavioural-variant FTD cohorts[73,102,104,105]; (ii) consistent with this proposal, in vivo human MRI studies have shown that the right dorsal STG and OFC are activated when normal subjects process social concepts[64–66]; (iii) there is direct evidence to suggest that the right dorsal STG and OFC have a crucial role in a disinhibition network and connect with other regions related to disinhibtion processing[106–109]. One benefit of using the regression and PCA methods is that it is possible to unpick the covariation of atrophy across frontotemporal regions in SD patients, and thus localise the small area of right dorsal STG and OFC compared with the widespread face-related area of the right temporal lobe. Consistent with this hypothesis, most very early right-sided SD cases (Table 3) were reported to behave entirely appropriately, with behavioural deficits emerging in some cases, perhaps when the atrophy has spread to or already encompassed the right dorsal STG.

The final dimension related to negative behaviours (i.e., apathy, depression and anxiety). Apathy is a principal symptom of behavioural-variant FTD[104,110]. As a variant of FTD, SD patients can also show apathy but typically less often[4,105]. We found that this factor was associated with those SD patients who had relatively more OFC damage. This result is consistent with previous studies which have shown a relationship between apathy and OFC in FTD patients[73,104]. We note that some previous investigations have observed more severe apathy in right > left SD patients[103]. Presumably, this might reflect the fact that, as found in the current clinical sample, right > left SD patients tend to have more atrophy of the ATL and OFC overall[62,70,84].

**Table 4 Review of the role of right dorsal superior temporal gyrus in social concepts processing.**

| First author | Year | Peak coordinate | Methods | Contrasts |
|---|---|---|---|---|
| Zahn | 2007 | 57 12 0 | fMRI | Social vs animal function concepts |
| Zahn | 2007 | 51 18 −12 | fMRI | Social vs animal function concepts corrected by social concepts vs fixation |
| Zahn | 2007 | 51 15 −12 | fMRI | Conjunction of social vs animal function concepts, correlations with descriptiveness of social behaviour and meaning relatedness |
| Zahn | 2009 | 54 0 −3 | fMRI | Effect of descriptiveness of social behaviour |
| Zahn | 2009 | 57 −18 6 | fMRI | Effect of descriptiveness of social behaviour |
| Zahn | 2009 | 60 −3 −3 | fMRI | Effect of descriptiveness of social behaviour masked by the same effect in Zahn et al.,[64] |
| Zahn | 2009 | 54 9 −24 | FTD | Hypometabolism of FTD with right superior ATL lesion |
| Zahn | 2009 | 51 9 −3 | FTD | Hypometabolism of FTD with right superior ATL lesion masked by Hypometabolism of FTD with social concept selective impairment |
| Ross | 2010 | 59 3 −19 | fMRI | Social vs neutral car video |
| Ross | 2010 | 66 −10 −24 | fMRI | Social vs animal function concepts |
| Pobric | 2016 | 53 8 −13 | TMS | Social vs non-social concepts |
| Binney | 2016 | 57 9 −18 | fMRI | Social vs matched-abstract concepts |
| Average | | 56 4 −10 | | |
| Our study | | 59 3 −2 | SD | Correlation of disinhibition and atrophy |
| Our study | | 51 6 −12 | SD | Correlation of disinhibition and atrophy |

The peak coordinates of our study are from the univariate voxel-based correlation mapping result.
*FTD* frontotemporal dementia, *TMS* transcranial magnetic stimulation, *SD* semantic dementia.

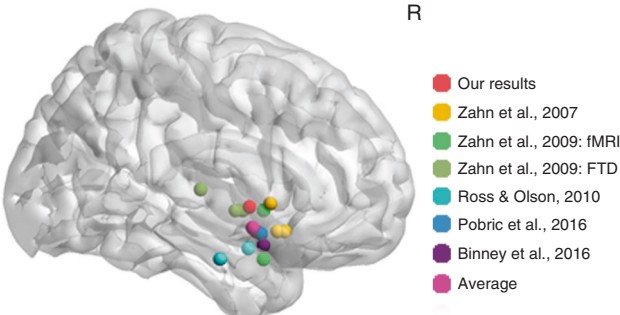

**Fig. 4** Review of the location of right dorsal superior temporal gyrus for social concepts processing from related literature and our study (the univariate voxel-based correlation mapping result). The brain map shows the peak locations of related studies. The right panel shows the specific names of these studies. FTD frontotemporal dementia.

From a methodological perspective, this study used a variety of analyses to relate cognitive-behavioural variation to the pattern of atrophy. Each has advantages and limitations. Although limited to selected ROIs, multiple regression (a multivariate approach) is better able than standard VBCM analysis to highlight when multiple regions contribute to a cognitive function. Thus, in this study the OFC bilaterally was linked to apathy and the ATL bilaterally to semantic and naming abilities (with the latter also linked to the left-right atrophy disparity). Secondly, we compared univariate and multivariate VBCM analyses (support vector regression/sparse canonical correlation analysis for neuroimaging; SVR/SCCAN) and found very similar results. These two approaches have complementary advantages and limitations[111,112]. Univariate analysis generates transparent and interpretable voxel weights but is limited by damage-colinearity, different variances and conservative multiple-comparison correction across voxels. In contrast, multivariate analysis considers all voxels simultaneously as a whole model and, thus multiple regions can be identified. However, the multivariate approach inevitably means that individual voxel weights are hard or impossible to interpret.

Finally, we note that there are some limitations with this research. First, only cross-sectional data were used. Future longitudinal studies will be able to capture the dynamic cognitive-

brain changes underpinning this neurodegenerative disease. Secondly, only unimodal neuroimaging data (grey matter volume) were used. Future multimodal neuroimaging studies, incorporating other neural information (e.g., local metabolism, temporal signals and region connectivity) should be able to provide a richer picture of the relationship between the patients' graded symptomology and the underlying neurological changes. Finally, we note that some types of behavioural problems such as obsessive behaviour and increased rigour can be missed by the NPI, which will be an important future development of the unified model of SD.

## Methods

**Participants**. Forty-seven dementia patients with prominent language problems (25 men; age: M = 63 years, s.d. = 7 years; range = 46–74 years; education level: M = 12 years; s.d. = 3 years, range = 3–18 years; years from onset: M = 3 years, s.d. = 2 years, range = 1–10 years; MMSE: M = 22, s.d. = 4; range = 13–29) were recruited from Huashan Hospital, Shanghai. They all met the diagnostic criteria for semantic-variant PPA[37]. According to the criteria, patients must present marked naming and single-word comprehension problems. Moreover, at least three of the features should be observed: the deficit of object knowledge, surface dyslexia, spared repetition and spared speech production. The exclusion criteria included a history of head trauma, neurological or psychiatric illness and a severe visuo-perceptual impairment. The structural MRI was used to support the diagnosis and exclude other potential comorbid conditions such as nondegenerative brain damage. All the patients presented atrophy in the ATL.

Twenty matched healthy participants (8 men, 12 women; age: M = 61 years, s.d. = 4 years; range = 51–69 years; education level: M = 10 years, s.d. = 3 years; range = 2–16 years) were selected as controls. They performed normally on the MMSE (M = 28, s.d. = 1, range = 26–30) with no history of neurological or psychiatric disorder.

All participants were right-handed native Chinese speakers, had normal or corrected-to-normal hearing and vision, and gave written informed consent. The study was approved by the Institutional Review Board of the Huashan Hospital, China.

**Assessments**. In oral picture naming, participants were instructed to provide the names of photographs presented on the screen. Object naming consisted of five categories (animals, non-manipulable objects, arbitrary artefacts, fruit and vegetables, and tools), each containing 20 items. The coloured photographs were used and the word frequency of items was matched between categories. Face naming was identical to the object naming, except only one category (20 photographs of famous faces) was presented. All the photographs of famous people were black-and-white. The famous people were Chinese actresses or actors, politicians and sport players.

In each trial of word-picture verification, a pair of picture and word was presented on the screen and participants were asked to judge whether they indicated the same object or famous person. In one condition the picture and word

matched. In the other, the word was replaced with the name of another exemplar from the same category. The match and mismatch conditions were counterbalanced across two assessment sessions with each item only appearing once per session. Only if participants answered correctly in both conditions (i.e., accepting the correct name and rejecting the semantic foil) was the item scored as correct. The stimuli and categories used in this test were the same as for picture naming, but there were only 10 items per category. The word frequency of items was also matched between five object categories.

Picture associative matching had the same format as the Pyramid and Palm Trees test[113], in which triplets of pictures were presented on the screen. Participants needed to choose which object/face was more semantically associated with the target. The stimuli and categories used in this test were the same as for picture naming. This test contained 60 trials, each category containing ten trials. The word frequency of the targets was matched between categories and the word frequency of answers and foils were matched within each object category.

Object perception task included 25 items, each in an array of three line drawings. The target was accompanied by two pictures from a different view. Participants were asked to match the pictures of the same object. The pictures in each trial were from the same category. Similarly, the face perception test required participants to judge whether two faces from different views were from the same person. Thirty-six items were included in this assessment. All the pictures were black-and-white photographs of male faces.

All the above assessments were from our home-made semantic battery but their psychometric properties have been systemically examined (e.g. sensitivity and specificity) among Chinese population. Moreover, they have been widely used in our previous studies about semantic processing on dementia, stroke patients and normal controls[114–116]. Assessments were administered using DMDX 4.0.1.0[117] in separate sessions in a fixed order to avoid the influence to pictures. Each session lasted no more than 2 h. Rest breaks were allowed. All participants completed this test battery, except one patient who did not finish the word-picture verification and three patients who did not finish the visual perception tasks. Comparisons between controls and patients were assessed using two-tailed independent two-sample $t$-tests by SPSS 13 (https://www.ibm.com/analytics/spss-statistics-software). The equality of variances for $t$-tests was tested by the Levene's test. If the assumption of equality of variances was rejected, the df would be corrected correspondingly.

The neuropsychiatric inventory questionnaire (NPI-Q)[118] was completed by the caregivers of 24 patients. They rated the presence and severity of 12 behaviour symptoms. Scores range from 0 to 3, representing absent, mild, moderate and severe changes, respectively. This version was the brief form of the NPI and its psychometric properties have been validated[119]. Because some symptoms included in the NPI were rarely endorsed for this patient group, we only chose six symptoms which occurred frequently. Therefore, symptoms of agitation, depression, anxiety, apathy, disinhibition and irritability were entered into our analyses. Chi-square tests were performed to compare the symptom severity between left and right SD groups. The NPI scores were transformed into negative $z$-scores, so that both the cognitive and behavioural scores ran in the same direction (low scores representing poor performance).

**MRI parameters and preprocessing procedure**. The 3D T1 images of all participants were collected through the Siemens 3 T scanner. The parameters were listed as follows: repetition time = 2300 ms, echo time = 2.98 ms, flip angle = 9°, matrix size = 240 * 256, field of view = 240 mm * 256 mm, slice number = 192 slices and voxel size = 1 mm * 1 mm * 1 mm.

The images were preprocessed with unified segmentation of SPM 12 (http://www.fil.ion.ucl.ac.uk/spm/). First, bias regularisation was conducted to remove the intensity inhomogeniety caused by the physics of MR scanning (light regularisation; bias FWHM = 60 mm). Then the images were segmented into grey matter (the number of Gaussians = 2), white matter (the number of Gaussians = 2), cerebrospinal fluid compartments (CSF; the number of Gaussians = 2), bone (the number of Gaussians = 3), soft tissue (the number of Gaussians = 4) and background (the number of Gaussians = 2) according to the tissue probability map of SPM. Next, they were normalised into the MNI space with both affine and non-linear transformations, resampled to 1.5 * 1.5 * 1.5 mm and modulated to compensate for the image warping during normalisation. Finally, they were smoothed with 8 mm FWHM.

**The atrophy of SD patients**. Voxel-based comparisons were employed between SD groups and controls using two-tailed independent $t$-tests (FDR-corrected $p < 0.05$) by REST 1.8 (http://www.restfmri.net/forum/). All the brain maps were made using BrainNet Viewer 1.63[120]. The voxels remaining in SD vs. controls were binarised as a mask for further voxel-based analyses.

As a variant of FTD, SD patients showed widespread frontotemporal atrophy[121]. To characterise the atrophy degree of temporal and frontal lobes, respectively, six related ROIs were derived from the Harvard-Oxford Atlas[122] (Fig. 1d). The bilateral ATL ROIs consisted of the anterior fusiform gyrus, anterior inferior temporal gyrus and temporal pole. The bilateral OFC ROIs were composed of the orbital frontal and subcallosal cortex. The bilateral hippocampus ROIs were composed of the hippocampus itself[123].

Four ROI measures were calculated. The difference between left and right ATL atrophy was generated by the grey matter volume of left ATL minus right ATL. The sum of ATL regions' grey matter volumes across hemispheres was used as the total ATL atrophy measure, the sum of bilateral OFC regions' grey matter volumes as the total OFC atrophy and the sum of bilateral hippocampus regions' grey matter volumes as the total hippocampus atrophy. Independent two-sample $t$-tests were implemented separately between SD groups vs. normal controls, and left vs. right SD patients by SPSS 13. In addition, the $z$-scores of patient's atrophy status across voxels were generated by regularising their grey matter volumes with respect to the control cohorts. The corresponding $z$-scores for the four ROI measures were extracted for subsequent analyses. Patients were divided into left > right and right > left groups according to the [left − right ATL] ROI measure.

**The relation between atrophy and task performance**. To understand how the laterality of temporal lobe atrophy, the total bilateral temporal and frontal lobe atrophy influenced patients' task performance, we built a series of linear regression models using SPSS 13, which used the four ROI $z$-scores as independent variables to predict all eight cognitive task measures and six NPI items separately. The $beta$ value for each independent variable was acquired by standardising all the independent and dependent variables to make their values comparable with each other. The significance of $beta$ values was tested by one sample $t$-tests compared with 0.

**The PCA of task performance**. To explore the underlying dimensions of variation in the patients' cognitive and behavioural measures, PCA analysis was implemented on the three face semantic, three object semantic, two visual perceptual tasks and six NPI items. Twenty-two subjects had missing data to some extent (ratio of missing data/available data = 19%), so we imputed their data to increase our statistical power for the PCA-related analysis. To achieve this, we first used the function 'pca_compsel' of PCA toolbox 1.3 (http://michem.disat.unimib.it/chm/) to determine the number of optimal factors of our data. Then the missing data were imputed by the function 'ppca' of matlab 2018b (https://www.mathworks.com/products/matlab.html). Finally, the imputed data were entered into the PCA by JMP 14 (https://www.jmp.com/en_us/home.html). All the scores were converted to $z$-scores based on the SD cohort before PCA. The factors with eigenvalue >1 were extracted and varimax rotation was applied to enhance cognitive interpretability of the principal components.

**The relation between atrophy and PCA factors**. To identify the relationship of temporal and frontal lobe status to the PCA factors, the corresponding ROI-based variables were again entered into the regression models using the same method as for the regions' models of specific task data (see above). The only exception here was that the dependent task variables were replaced by PCA factor scores.

To explore whether other regions beyond the ROIs also associated with the patients' PCA scores, we completed VBCM analyses between the voxel-based atrophy $z$-scores and each PCA factor within the atrophy mask. Both uni- and multivariate analyses were conducted.

For the univariate analysis, two-tailed correlation analysis was conducted between each voxel's atrophy $z$-score and each factor score using REST 1.8. We adopted continuous permutation-based FWER correction for multiple comparisons[124]. This method is stricter than FDR, provides greater flexibility on cluster size and is more transparent. Here we used the threshold of $v = 100$ that represents the number of false positive voxels allowed.

Two multivariate algorithms were performed using LESYMAP 0.0.0.9221[125]. Before the analysis, all atrophy data were scaled across patients to keep the same scale as factor scores. For SCCAN, we chose a directional model, allowing both positive and negative weightings. The sparseness was optimised to obtain the best model. To display crucial regions, only voxels with expected direction and weights >0.1 were presented. For SVR, a radial epsilon kernel was adopted with gamma = 5 and cost = 30. The voxel statistical inference was generated by comparing the voxel's real $beta$ value with its 1000-times permuted $beta$ values[126]. Negative and positive $beta$ values were considered separately. All the SCCAN and SVR models' effects were evaluated using 4-fold cross-validation.

**Face ROI analysis**. Face processing is associated with widespread right temporal areas[98], thus we explored which regions were involved in SD's face deficits. Five classic face-related ROIs were chosen from previous literature (the Talairach coordinates of occipital face area: 25, −88, −10; fusiform face area: 40, −44, −16; posterior superior temporal sulcus: 48, −48, 8; ventral ATL: 25, 0, −28; anterior superior temporal sulcus: 48, −12, −7)[127]. We transformed these Talarich coordinates into the MNI space and generated 6-mm spheres for all the ROIs. Then, we extracted the atrophy $z$-scores of these ROIs and correlated them with the face-related PCA scores.

To validate the specific role of these right-sided ROIs in face processing (over the left hemisphere homologues), supplementary analyses were conducted to (1) correlate the right ROI atrophy $z$-scores with object processing tasks; (2) correlate the left corresponding ROI status with face processing scores; and (3) correlate the right ROI volumes in the control group against face processing performance.

**Reporting summary**. Further information on research design is available in the Nature Research Reporting Summary linked to this article.

## Data availability

All relevant data including task scores, grey matter volume images and Harvard-Oxford atlas are available by request to the authors. Source data underlying Figs. 1–3, Table 2 and Supplementary Tables 1–6 are provided as a Source data file. A reporting summary for this Article is available as a Supplementary Information file.

## Code availability

All relevant codes are available by request to the authors.

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

## Acknowledgements

We remain grateful to the patients and their carers for supporting this research. We thank Dr. Ajay Halai for numerous helpful discussion and processing scripts. This work was supported by the Chinese Scholarship Council to J.D. and H.L.; Medical Research Council programme grant and intramural funding (MR/R023883/1 and MC_UU_00005/18) and ERC grant (GAP: 670428 - BRAIN2MIND_NEUROCOMP) to MALR; National Natural Science Foundation of China (31872785 & 81171019) and Beijing Natural Science Foundation (7182088) to Z.H.; National Key R&D Program of China (2016YFC1306305) to Q.G.

## Author contributions

The study was proposed and supervised by M.A.L.R., Q.G. and Z.H. The semantic tests were designed by Z.H. The patients were diagnosed by Q.G. The behavioural and imaging data were collected by K.C., Q.Y., L.H. and Y.L. The behaviour comparison and PCA were carried out by H.L. and Y.C. The brain-behaviour analysis was carried out by J.D., K.C. and M.A.L.R. The paper was written by M.A.L.R. and J.D. The figures and tables were made by M.A.L.R., J.D. and H.L.

## Competing interests

The authors declare no competing interests.
