## [Peer Review File · Nature Communications]

Reviewers' comments:

Reviewer #1 (Remarks to the Author):

Ding et al report correlations between behavioral/cognitive measures and volume loss in a consecutive series of semantic dementia patients. PCA revealed 5 principal components, two of which summarize behavioral disturbances.

The sample size of 47 is relatively large and appropriate for a rare disease, the analysis is sound. The main results are based on voxel-based mapping (Fig 2). They confirm the role of medial prefrontal cortex in apathy (F1). The network underlying face and object associative matching confirms the role of the right anterior temporal pole and shows additional involvement of ventromedial prefrontal cortex as well as lateral occipitotemporal regions. The latter was further examined based on face-processing ROIs from the literature (Fig 3). The results confirm the critical role of the left anterior temporal pole in word comprehension, object naming and naming of familiar faces. Factors F4 and F5 gave less clear atrophy correlates (Fig 2). The data are then supplemented by a review of the literature of right-sided temporal lobe atrophy. Overall, this systematic study of a relatively large group of SD cases is very well conducted, of significant interest and of high clinical relevance. Results mainly confirm prior findings.

Minor comment

1. P 24: The analogy between the variation in patient presentations and the variety of bakery products may be theoretically insightful, it runs throughout the discussion but could be felt to be disrespectful to patients. Furthermore, the analogy misses sth quite substantial: If one analyses the composition of food chemically, all ingredients will be identified, even those one may not expect, however in the current study design, only those components will be found that are measured by the a priori selected set of tests.

Reviewer #2 (Remarks to the Author):

In this manuscript, Ding et al. aimed to develop an integrative, unified model to map the heterogeneous clinical presentations of semantic dementia (SD) to the patient's distribution of frontotemporal atrophy. Specifically, the authors used a mapping approach to examine variations in cognitive and behavioural presentations in SD in association with brain atrophy in the anterior temporal lobes (ATLs) and other neighbouring cortical regions (insula, orbitofrontal cortex; OFC). The patient cohort included a relatively large sample (being SD a rare syndrome) of both left and right-predominant SD patients. Results from regression analyses associated the degree of semantic impairment with bilateral ATL atrophy. The authors further identify with PCA 5 clinical dimensions associated with unique patterns of cortical atrophy, including cortical regions other than the ATL.

This manuscript has several strengths. First, it should be noted that the senior author of this work is an internationally recognised cognitive neuroscientist with significant experience investigating the cognitive and neural basis of conceptual knowledge. The study is well designed and the manuscript is very well

written. The introduction in particular does a great job summarising what is currently known about semantic processing in SD, integrating findings from healthy and clinical populations. The methods and statistical analyses are sound and the conclusion summarises well the findings and integrates them with the literature.

The weaker aspects of this manuscript include the cross-sectional approach, the unimodal (structural) MRI analyses and the questions regarding the novelty of the MRI methods and results. The scope of the manuscript is unclear. The clinical significance of the findings is not discussed and there are no directions/suggestions for future research or consideration of caveats/limitations. I have also minor comments concerning aspects of the imaging methods.

I was somewhat uncertain of the scope of this manuscript (cognitive versus clinical?). In the first part of the manuscript, the authors develop a model of the contributions of ATL to cognition and behaviour using SD as a lesion model. Then, the generalisation of the model is tested using longitudinal case reports of right-SD in the literature with an unclear purpose (disease staging? accurate diagnosis/prognosis?).

SD is a neurodegenerative disorder. As the disease progresses brain atrophy encroaches into new brain regions, which in turn cause the emergence of clinical symptoms. Importantly, these changes will also occur across different structural and functional brain properties. From a clinical perspective, cross-sectional, unimodal investigations/modelling are therefore unable to capture this dynamic process and in turn can only present a limited snapshot of the disease. As such claims of progression of changes need to be revised. From a theoretical perspective, unimodal VBM MRI studies ignore the contribution of other brain changes (functional, connectivity) to the emergence of diverse cognitive and clinical profiles. There, claims of structural brain atrophy in discrete regions reflecting cognitive/clinical performance should be tempered down.

The model is developed in an elegant way using a two stage approach; regression analysis and PCA (and further post-hoc generalisation). Quantitative brain mapping techniques, including regression analysis and PCA, however, are hardly novel. This is acknowledged by the authors themselves in the introduction. These are well established approaches that have been used before in other brain disorders, including development of diagnostic criteria for PCA. I wonder whether novel graded neuroimaging analyses, which the senior author is familiar with (i.e., Bajada et al., Cortex 2019), would be better suited to investigate how graded clinical differences in patients map to the spatial distribution of frontotemporal atrophy?

Although whole-brain regression analyses are performed, the study uses primarily an atlas-based ROI approach. Indeed, brain behaviour associations are examined using 3 ROI-derived measurements. ROIs can be derived using various approaches (i.e., masking out whole-brain data-driven maps of atrophy in patients versus controls, using cluster information to identify a hierarchy of regions based on the peak values, etc.). The atlas-based, a priori selection of discrete regional boundaries in this study, however, risks missing out the implication of brain regions other than the bilateral ATL and OFC (i.e., asymmetric hippocampal involvement is considered one of the hallmark features of SD).

I also wonder whether the findings add significantly to the literature? It seems the model largely replicate, albeit in a single, unified analysis, what other studies have reported before, in terms of the contribution of cortical regions to the clinical/cognitive deficits. What are the novel insights?

Minor points:

There are typos in the legend of figure 3 and on the title of table 3.

Please add left and right labels in figures. What is T representing in the figure? What is the measure of the thresholds (i.e., log10 of p value?)

How many cases had missing data? What is the proportion of imputed vs available data?

Factor 5 in the PCA (12% of variance explained) had high loadings for object and face perception. What is the rationale for labelling this factor "disease severity"?

I would suggest changing 'gender' for 'sex'. Neither are binary dimensions but 'sex' is arguably more closely related to the biology.

Is there a measure of disease duration or any other measure of disease severity (i.e., everyday functional performance) for patients?

Reviewer #3 (Remarks to the Author):

This manuscript reports an investigation of a relatively large (N=47) and heterogeneous group of FTD patients. The assessments included behavioral/cognitive measures of naming, face and object processing, neuropsychiatric symptoms (symptoms of agitation, depression, anxiety, apathy, disinhibition, and irritability), and MRI to assess the precise pattern of neurodegeneration. The key contribution here is the consideration of multidimensional cognitive and behavioral-emotional deficits, and how they relate to different neural progression of the disease. I am broadly sympathetic to the theoretical position(s) of the authors and, as far as I can tell, none of their claims are particularly controversial. However, bringing together these different strains of cognitive and clinical neuroscience research is an important contribution. PCA and voxel-based analyses are a useful approach to this and there is a small systematic review of clinical cases and of studies on social concepts that make useful additions.

The main weakness of the study is that the neural evidence appears not to be very strong. The ROI analyses are (necessarily, deliberately) coarse and mostly relate to anomia, which doesn't really capture the multidimensionality that is the main strength of this study. Anomia is also a very well-studied symptom, so that contribution on its own is not particularly major. The voxel-based analyses are potentially much more informative as they reveal important differences in the neural correlates of the

deficit “ingredients” (clever analogy, that). However, from the methods section, it seemed to me that there was no correction for multiple comparisons in the VBCM, or possibly FDR correction. Lack of correction for multiple comparisons would be a serious problem and would allow, at best, qualitative hypothesis generation for future studies. FDR correction is better, though it would likely be anti-conservative (Mirman et al., 2018, *Neuropsychologia*, 115, 112-123), which would also weaken any inferences. A non-parametric permutation-based FWER control method would be best and I recommend re-running the VCBM using that approach. (Of course, if I misunderstood and the authors already did that, then it simply needs some clarification).

A multivariate VCBM analysis approach such as SVR or SCCAN would also be more cutting-edge, but it is not strictly necessary.

It might be worthwhile to connect this study with work by Behrmann & Plaut (2013, *Trends Cogn Sci*, 17:5, 210-219) in which a single set of computational principles leads to development of lateralized VWFA and FFA. The argument here seems similar -- both ATLs perform a similar computational role, but proximity to left-lateralized language areas lead to left ATL (relative) specialization for verbal semantics (esp. naming) and right ATL (relative) specialization for face/social semantics.

Reviewer #1 (Remarks to the Author):

Ding et al report correlations between behavioral/cognitive measures and volume loss in a consecutive series of semantic dementia patients. PCA revealed 5 principal components, two of which summarize behavioral disturbances.

The sample size of 47 is relatively large and appropriate for a rare disease, the analysis is sound. The main results are based on voxel-based mapping (Fig 2). They confirm the role of medial prefrontal cortex in apathy (F1). The network underlying face and object associative matching confirms the role of the right anterior temporal pole and shows additional involvement of ventromedial prefrontal cortex as well as lateral occipitotemporal regions. The latter was further examined based on face-processing ROIs from the literature (Fig 3). The results confirm the critical role of the left anterior temporal pole in word comprehension, object naming and naming of familiar faces. Factors F4 and F5 gave less clear atrophy correlates (Fig 2). The data are then supplemented by a review of the literature of right-sided temporal lobe atrophy. Overall, this systematic study of a relatively large group of SD cases is very well conducted, of significant interest and of high clinical relevance.

Response: We thank R1 for these very positive comments. These are much appreciated given the considerable effort involved in recruiting this patient sample and to undertake the various existing and new analyses.

Minor comment

1. P 24: The analogy between the variation in patient presentations and the variety of bakery products may be theoretically insightful, it runs throughout the discussion but could be felt to be disrespectful to patients. Furthermore, the analogy misses sth quite substantial: If one analyses the composition of food chemically, all ingredients will be identified, even those one may not expect, however in the current study design, only those components will be found that are measured by the a priori selected set of tests.

Response: Thank you for these two thoughtful comments. With regard to the analogy itself; of course, we are deeply respectful of all patients including those who are kind enough to participate in our studies (cf. the acknowledgement section in all our papers). And we are not, of course, trying to suggest that patients and their challenging symptoms are baked goods. We have found that, when presenting these results in the past, academic and non-academic audiences sometimes find it harder to understand what PCA results reflect, but find this baking analogy particularly helpful. Accordingly we have reviewed and revised the text such that it is crystal clear that this is an analogy related to the PCA itself and not about the patients.

The second point is very well made. We entirely agree that PCA and multiple regressions are limited to the measures included. (Indeed, one could note that chemical spectroscopy is also limited by its ability to distinguish different molecular frequencies, and thus compounds with similar frequencies can also mask each other.) Accordingly, in the revised text, we acknowledge that the PCA components are limited to the assessments included (see page 24, lines 16-18).

Reviewer #2 (Remarks to the Author):

In this manuscript, Ding et al. aimed to develop an integrative, unified model to map the heterogeneous clinical presentations of semantic dementia (SD) to the patient's distribution of frontotemporal atrophy. Specifically, the authors used a mapping approach to examine variations in cognitive and behavioural presentations in SD in association with brain atrophy in the anterior temporal lobes (ATLs) and other neighbouring cortical regions (insula, orbitofrontal cortex; OFC). The patient cohort included a relatively large sample (being SD a rare syndrome) of both left and right-predominant SD patients. Results from regression analyses associated the degree of semantic impairment with bilateral ATL atrophy. The authors further identify with PCA 5 clinical dimensions associated with unique patterns of cortical atrophy, including cortical regions other than the ATL.

This manuscript has several strengths. First, it should be noted that the senior author of this work is an internationally recognised cognitive neuroscientist with significant experience investigating the cognitive and neural basis of conceptual knowledge. The study is well designed and the manuscript is very well written. The introduction in particular does a great job summarising what is currently known about semantic processing in SD, integrating findings from healthy and clinical populations. The methods and statistical analyses are sound and the conclusion summarises well the findings and integrates them with the literature.

Response: As per R1, we thank R2 for these very positive comments and evaluation of our study. Recruiting this size of SD patients and undertaking the assessments and analyses are logistically challenging – thus these comments are very much appreciated.

The weaker aspects of this manuscript include the cross-sectional approach, the unimodal (structural) MRI analyses and the questions regarding the novelty of the MRI methods and results. The scope of the manuscript is unclear. The clinical

significance of the findings is not discussed and there are no directions/suggestions for future research or consideration of caveats/limitations. I have also minor comments concerning aspects of the imaging methods.

Response: Thank you for these important areas, highlighted for improvement. These suggestions are laid out in more detail below, alongside our response to each of them.

I was somewhat uncertain of the scope of this manuscript (cognitive versus clinical?). In the first part of the manuscript, the authors develop a model of the contributions of ATL to cognition and behaviour using SD as a lesion model. Then, the generalisation of the model is tested using longitudinal case reports of right-SD in the literature with an unclear purpose (disease staging? accurate diagnosis/prognosis?).

Response: As R2 is probably aware, we are committed to both cognitive and clinical explorations. Indeed, we are particularly keen to develop theories and mechanistic accounts that simultaneously explain both healthy and impaired function. Such convergent evidence is a powerful approach to cognitive-clinical neuroscience. Thus in the revised manuscript (in the Abstract, Introduction and Discussion) we have made it clear that we are striving for a model that can explain SD data but also results from other patient groups and healthy participants. Because the prominent role of the ATL in semantic function was founded, in large part, upon the unique insights from SD patients then this paper focusses primarily on revisiting the variations of behavior across this patient group. We then consider how this emergent model fits with the other forms of data in the Discussion.

As noted by R2, in this study we undertake both a detailed examination of an SD cohort and then consider how this relates to published data from early SD patients with right-sided temporal lobe atrophy. As we make clear in the revised manuscript – our clinical SD sample is very much like those reported previously by other major international research groups. For reasons that are still not clear, most SD patients present clinically with early left-sided or bilateral ATL atrophy. In contrast early-presenting right-sided cases are much rarer. This pattern can be seen in the atrophy scatterplots in Figure 1. Thus in the first main part of the paper we develop a multi-dimensional model using these data. The second part then tests an extrapolation of this model into the unpopulated areas of the underpinning data – namely, patients with early, mild, right-sided atrophy. Given that, by definition, such patients are not available within the same sample – we revisited all of the single cases that we could locate within the literature and found that they fitted well with the extrapolation derived by the model.

Taken together these results provide insights and new ways of thinking about both basic cognitive and clinical neuroscience matters. These include the notion that the ATLs work in tandem to support core semantic functions but there are graded differences in their contribution to speech production and face recognition depending on the distribution of atrophy. At the same time, from a clinical perspective, rather than a never-ending splitting of this patient group into multiple “subtypes or categories” (when in fact there are no absolute boundaries within or between patients), the results show how the graded variations across patients can be captured by a multidimensional rather than multi-categorical approach. Thus, as per our overarching aim, the investigation provides important results for both basic cognitive and clinical neuroscience. These strategic factors and aims are considered in the Introduction and Discussion of the revised manuscript.

SD is a neurodegenerative disorder. As the disease progresses brain atrophy encroach into new brain regions, which in turn cause the emergence of clinical symptoms. Importantly, these changes will also occur across different structural and functional brain properties. From a clinical perspective, cross-sectional, unimodal investigations/modelling are therefore unable to capture this dynamic process and in turn can only present a limited snapshot of the disease. As such claims of progression of changes need to be revised. From a theoretical perspective, unimodal VBM MRI studies ignore the contribution of other brain changes (functional, connectivity) to the emergence of diverse cognitive and clinical profiles. There, claims of structural brain atrophy in discrete regions reflecting cognitive/clinical performance should be tempered down.

Response: We appreciate these helpful suggestions. We agree, of course, that SD is a progressive neurodegenerative disease, meaning that clinical symptoms can emerge through increased atrophy within already affected regions and/or also encroachment into new (typically neighbouring or connected) areas. Accordingly, we have added a limitations section to the Discussion to acknowledge these points (see page 30 , lines 15-22). And, as noted below, we have also repeated the symptom-atrophy analyses using two multivariate methods as well as the existing univariate approach and the (multivariate) multiple regressions.

The model is developed in an elegant way using a two stage approach; regression analysis and PCA (and further post-hoc generalisation). Quantitative brain mapping techniques, including regression analysis and PCA, however, are hardly novel. This is acknowledged by the authors themselves in the introduction. These are well

established approaches that have been used before in other brain disorders, including development of diagnostic criteria for PCA. I wonder whether novel graded neuroimaging analyses, which the senior author is familiar with (i.e., Bajada et al., Cortex 2019), would be better suited to investigate how graded clinical differences in patients map to the spatial distribution of frontotemporal atrophy?

Response: As noted above, in the new manuscript, we have added two multivariate VBCM analyses in addition to the univariate voxel-based and multiple regression ROI-based investigations. Importantly the univariate and multivariate produce very similar outcomes for the extracted behavioural factors (see page 38, lines 16-21; page 39, lines 1-2; fig 2). Interestingly a recently-published investigation (Schumacher et al, Brain 2019) also found that very similar results were found with multivariate and univariate analyses in post-stroke aphasia.

Although whole-brain regression analyses are performed, the study uses primarily an atlas-based ROI approach. Indeed, brain behaviour associations are examined using 3 ROI-derived measurements. ROIs can be derived using various approaches (i.e., masking out whole-brain data-driven maps of atrophy in patients versus controls, using cluster information to identify a hierarchy of regions based on the peak values, etc.). The atlas-based, a priori selection of discrete regional boundaries in this study, however, risk missing out the implication of brain regions other than the bilateral ATL and OFC (i.e, asymmetric hippocampal involvement is considered one of the hallmark features of SD).

Response: Of course, we agree with R2 about the limitations of ROI-based examinations. This is why we considered it important to include both ROI and whole-brain analyses. Following R2's observation that more priori regions should be included, especially the hippocampus (La joie et al., 2014), we added this region into the ROI regression analysis. We found that the sum of the hippocampus was negatively related with the anxiety.

I also wonder whether the findings add significantly to the literature? It seems the model largely replicate, albeit in a single, unified analysis, what other studies have reported before, in terms of the contribution of cortical regions to the clinical/cognitive deficits. What are the novel insights?

Response: Whilst we agree that each 'symptom' of SD covered in the study is not new (we are not revealing new symptoms for the first time – indeed SD is perhaps one of the best cognitively-investigated patient groups), as far as we are aware no one has

tried to generate a unified model of SD that can assimilate all these features and explain their variations across patients. As noted above, such a model allows us to provide insights about the basic cognitive science of the bilateral ATL system and also, clinically, to explain how the graded variations of symptom across SD patients can be understood and recognized (which itself is important for accurate diagnosis and to ensure that relevant patients are not excluded) (see page 3, lines 9-21).

Minor points:

There are typos in the legend of figure 3 and on the title of table 3.

Response: *Done. Figure 3: PCA scores → face factor scores (see figure 3); Table 3: voxel-wised → voxel-wise (see supplementary table 1)*

Please add left and right labels in figures. What is T representing in the figure? What is the measure of the thresholds (i.e., log₁₀ of p value?)

Response: *Done. T means t values. The thresholds are the FDR corrected $p < 0.05$ (see Figure 1).*

How many cases had missing data? What is the proportion of imputed vs available data?

Response: *Twenty-two subjects had missing data. The ratio of missing data/available data = 19% (see page 37, line 9-10).*

Factor 5 in the PCA (12% of variance explained) had high loadings for object and face perception. What is the rationale for labelling this factor “disease severity”?

Response: *This is because perception impairments only occurred in the very late stage of SD, we refer to this factor as ‘SD severity’, an index to distinguish severe and mild SD patients (see page 16, lines 17-18).*

I would suggest changing ‘gender’ for ‘sex’. Neither are binary dimensions but ‘sex’ is arguably more closely related to the biology.

Response: *Done (page 9, line 10).*

Is there a measure of disease duration or any other measure of disease severity (i.e., everyday functional performance) for patients?

Response: *Yes, we added the disease duration information from patients’ own reports. The SD group, left- and right sided subgroups had similar durations ($M = 3$ years, $s.d. = 2$ years). Left- and right- sided subgroups have no significant difference on this variable ($t = 0.69$, $p = 0.50$; see table 1).*

Reviewer #3 (Remarks to the Author):

This manuscript reports an investigation of a relatively large (N=47) and heterogeneous group of FTD patients. The assessments included behavioral/cognitive measures of naming, face and object processing, neuropsychiatric symptoms (symptoms of agitation, depression, anxiety, apathy, disinhibition, and irritability), and MRI to assess the precise pattern of neurodegeneration. The key contribution here is the consideration of multidimensional cognitive and behavioral-emotional deficits, and how they relate to different neural progression of the disease. I am broadly sympathetic to the theoretical position(s) of the authors and, as far as I can tell, none of their claims are particularly controversial. However, bringing together these different strains of cognitive and clinical neuroscience research is an important contribution. PCA and voxel-based analyses are a useful approach to this and there is a small systematic review of clinical cases and of studies on social concepts that make useful additions.

Response: again we thank R3, like R1 and R2 about this very positive evaluation of our study.

The main weakness of the study is that the neural evidence appears not to be very strong. The ROI analyses are (necessarily, deliberately) coarse and mostly relate to anomia, which doesn't really capture the multidimensionality that is the main strength of this study. Anomia is also a very well-studied symptom, so that contribution on its own is not particularly major.

Response: We note R3's opinion which is less optimistic than that from R1 and R2. We note that (a) the study uses more than just ROI analyses, and – following suggestions from R2 and R3 (below), now also includes multivariate as well as univariate whole-brain analyses And (b) these analyses were able to show formally the multidimensionality in the behavioural data through the PCA (which we believe is the first time this has been undertaken in SD – though has been done in post-stroke aphasia) and that these link to various brain regions. Thus, for example, overall semantic and comprehension performance relates to the degree of bilateral atrophy, anomia to bilateral load plus the ATL asymmetry, and apathy to OFC atrophy. Given that we are exploring a single, relatively homogeneous patient group with graded behavioural variations (rather than a large undifferentiated lesion mapping study of

different aetiologies and areas of brain damage), we feel – like R1 and R2 – that the study provides important new approaches and results.

The voxel-based analyses are potentially much more informative as they reveal important differences in the neural correlates of the deficit “ingredients” (clever analogy, that). However, from the methods section, it seemed to me that there was no correction for multiple comparisons in the VBCM, or possibly FDR correction. Lack of correction for multiple comparisons would be a serious problem and would allow, at best, qualitative hypothesis generation for future studies. FDR correction is better, though it would likely be anti-conservative (Mirman et al., 2018, *Neuropsychologia*, 115, 112-123), which would also weaken any inferences. A non-parametric permutation-based FWER control method would be best and I recommend re-running the VCBM using that approach. (Of course, if I misunderstood and the authors already did that, then it simply needs some clarification).

A multivariate VCBM analysis approach such as SVR or SCCAN would also be more cutting-edge, but it is not strictly necessary.

Response: We really appreciate this wise and helpful suggestion. Therefore, we have updated our VCBM results. First, univariate VCBM is now corrected by the permutation-based FWER (see page 38, lines 12-15). Second, multivariate VCBM (SCCAN) was also conducted by LESYMAP (see page 38, lines 16-21; page 39, lines 1-2).

The uni- and multi-variate VCBM generated very similar results, confirming the reliability of our results. We have briefly discussed their pros and cons in the Discussion (see page 30, lines 7-14).

It might be worthwhile to connect this study with work by Behrmann & Plaut (2013, *Trends Cogn Sci*, 17:5, 210-219) in which a single set of computational principles leads to development of lateralized VWFA and FFA. The argument here seems similar -- both ATLs perform a similar computational role, but proximity to left-lateralized language areas lead to left ATL (relative) specialization for verbal semantics (esp. naming) and right ATL (relative) specialization for face/social semantics.

Response: Thanks for recommending this nice paper to us. Indeed, this framework parallels our own theoretical work on the ATL (stemming back to Lambon Ralph et al., 2001). This work is highly related to our research in many aspects. Therefore we have cited it (see page 26, line 19)

References

Schumacher, R., Halai, A. D., & Lambon Ralph, M. A. (2019). Assessing and mapping language, attention and executive multidimensional deficits in stroke aphasia. Brain, 142(10), 3202-3216.

La Joie, R., Landeau, B., Perrotin, A., Bejanin, A., Egret, S., Pélerin, A., ... & Desgranges, B. (2014). Intrinsic connectivity identifies the hippocampus as a main crossroad between Alzheimer's and semantic dementia-targeted networks. Neuron, 81(6), 1417-1428.

****REVIEWERS' COMMENTS:**

Reviewer #1 (Remarks to the Author):

Major comments

1. For the reasons mentioned in my previous review, I insist that the sentence 'variation in patient presentations is like the array of breads, cakes and patisseries found in a bakery, with this diversity reflecting the differing amounts of the key ingredients (flour,butter, eggs, etc.).' is removed.
2. An important limitation is that the assessment of behavioral changes is limited to those detected by the NPI. The NPI was developed to test AD but does not capture the behavioral problems in SD properly. For instance, a frequent behavioral change in SD is obsessive behavior, with repetitiveness and also increased rigor. This is not captured by the NPI but is one of the key dimensions in SD.

Reviewer #2 (Remarks to the Author):

The authors have comprehensively addressed all the reviewers' concerns. To this reviewer, the manuscript is ready for publication.

Reviewer #3 (Remarks to the Author):

The authors have effectively addressed my critiques of their initial submission. The original submission was impressive and important because of its scope and strong integration of data-driven and theory-driven approaches. My main concern was the neural evidence did not seem very strong, but the addition of multiple comparisons correction and multivariate LSM has resolved that problem. The convergence across these different analysis methods indicates a fairly robust pattern that I believe will be an important contribution to the field.

Reviewer #1 (Remarks to the Author):

Major comments

1. For the reasons mentioned in my previous review, I insist that the sentence ‘variation in patient presentations is like the array of breads, cakes and patisseries found in a bakery, with this diversity reflecting the differing amounts of the key ingredients (flour,butter, eggs, etc.).’ is removed.

Response: We thank for R1’s kind reminder. Therefore, we have removed this sentence.

2. An important limitation is that the assessment of behavioral changes is limited to those detected by the NPI. The NPI was developed to test AD but does not capture the behavioral problems in SD properly. For instance, a frequent behavioral change in SD is obsessive behavior, with repetitiveness and also increased rigor. This is not captured by the NPI but is one of the key dimensions in SD.

Response: We appreciate this comment. This limitation has been mentioned in the revision (see page 21, lines 2-4).